

# Rapid decline of Arctic sea ice volume: Causes and consequences
Jean-Claude Gascard (1), Jinlun Zhang (2) and Mehrad Rafizadeh (1)
(1) LOCEAN, Sorbonne Université, Paris, France
(2) Polar Science Center, Applied Physics Lab, University of Washington, Seattle, USA
## **Abstract.**
The drastic reduction of the Arctic sea ice over the past 40 years is the most glaring evidence of
climate change on Planet Earth. Among all the variables characterizing sea ice, the sea ice volume is
by far the most sensitive one for climate change since it is decaying at the highest rate compared to
sea ice extent and sea ice thickness. In 40 years the Arctic Ocean has lost about 3/4 of its sea ice
volume at the end of the summer season corresponding to a reduction of both sea ice extent and sea
ice thickness by half on average. From more than 16000 km$^3$, 40 years ago, the Arctic sea ice summer
minimum dropped down to less than 4000 km$^3$ during the most recent summers. Being a
combination of Arctic sea ice extent and sea ice thickness, the Arctic sea ice volume is difficult to
observe directly and accurately. We estimated cumulative Freezing-Degree Days (FDD) over a 9
month freezing time period (September to May each year) based on ERA Interim surface air
temperature reanalysis over the whole Arctic Ocean and for the past 38 years. Then we compared
the Arctic sea ice volume based on sea ice thickness deduced from cumulative FDD with Arctic sea ice
volume estimated from PIOMAS (Pan Arctic Ice Ocean Modeling and Assimilation System) and from
the ESA CRYOSAT-2 satellite. The results are strikingly similar. The warming of the atmosphere is
playing an important role in contributing to the Arctic sea ice volume decrease during the whole
freezing season (September to May). In addition, the FDD spatial distribution exhibiting a sharp
double peak-like feature is reflecting the Multi Year Ice (MYI) versus First Year Ice (FYI) dual
disposition typical of the Arctic sea ice cover. This is indicative of a significant contribution from the
vertical ocean heat fluxes throughout the ice depending on MYI versus FYI distribution and the snow
layer on top of it influencing the surface air temperature accordingly. In 2018 the Arctic MYI vanished
almost completely for the first time ever over the past 40 years. The quasi complete disappearance
of the Arctic sea ice is more likely to happen in summer within the next 15 years with broad
consequences for Arctic marine and terrestrial ecosystems, climate and weather patterns on a
planetary scale and globally on human activities.

## **1/ Introduction**
It is well recognized that the Arctic Sea Ice extent and thickness decreased drastically over the past
40 years as shown by Earth observing satellites and as reported extensively by many scientists. Over
the past 20 years the amount of scientific publications regarding Arctic sea ice evolution and
behavior over hours to decades at local, regional, and pan-Arctic scales, is exceptional. This is so
because among several major elements of the Earth climate system, the actual Arctic sea ice decline
is one of the most representative characteristics of climate change. In the past, the main aspect



concerned Arctic sea ice extent largely based on space observations. For instance **Serreze et al.**
[2003] reported a "record minimum" for sea ice extent in 2002 followed by **Stroeve et al.** [2005] who
reported "another extreme minimum" in 2004.  **Comiso** [2006] described "an abrupt decline in the
Arctic winter sea ice cover in 2005 and 2006 and **Kwok** [2007] "a near zero replenishment of the MYI
at the end of the 2005 summer". Then came the exceptional summer of 2007 during the 4[th]
International Polar Year (IPY, 2007-2008) characterized by a phenomenal Arctic sea ice extent
reduction never observed before during the satellite era. Thanks to the IPY stimulating a major effort
from the scientific community, the first decade of the 21st century ended with an unprecedented
amount of new results regarding the Arctic sea ice (**Giles et al.** [2008], **Zhang et al.** [2008], **Perovich**
et al. [2008], **Kauker et al.** [2009] to name a few). Since then, 10 years have passed and sea ice extent
and thickness have further decreased.  **Zhao et al.** (2018) described a strong decrease of sea ice
concentration in the entire central Arctic in 2010. The whole time record for summer sea ice extent
minimum was reached in September 2012 (3.4 x 10$^6$ km$^2$) and sea ice volume (3800 km$^3$). More
recently **Stroeve et al.** (2018) stressed our attention about a sharp drop of sea ice thickness occurring
in 2015-2016. This continuous chain of events maintained a strong motivation among scientists for
Arctic sea ice research both from a modeling and experimental point of view taking advantage of
new technologies for observations and more sophisticated models. In addition there is now a great
interest expressed by meteorologists due to very peculiar and intriguing winters occurring in the
northern hemisphere since 2015 (**Moore** [2016], **Cullather et al.** [2016], **Binder et al.** [2017]). This
"peculiarity"  is mainly characterized by a large scale atmospheric circulation and extra-tropical
cyclones bringing warm air masses to the North Pole and cold air outbreaks impacting mid latitudes
regions all related to large scale Arctic sea ice variability (**Overland and Wang** [2010], **Tang et al.**
[2013], **Rinke et al.** [2017], **Kim et al.** [2017], **Graham et al.** [2017]).
Most of the results obtained so far concerned Arctic sea ice extent and Arctic sea ice thickness taken
separately. In order to better analyze and understand Arctic sea ice evolution, an important step was
accomplished by introducing Arctic sea ice volume. Sea Ice volume is an important parameter
although very challenging to estimate precisely since it is a combination of sea ice area and sea ice
thickness. Sea ice volume is decreasing at a much higher rate than sea ice extent and sea ice
thickness which explains the greater sensitivity of sea ice volume to characterize climate change. As
shown in the following, Arctic sea ice volume has decreased by as much as 75% at the end of the
summer season if compared with the situation 40 years ago (from more than 16000 km$^3$ in the late
70s it was less than 4000 km$^3$ in September 2012). In contrast Arctic sea ice extent and thickness
have both decreased by half during the same time period accordingly.
In this paper we will revisit the whole time period extending from 1979 until now by estimating and
comparing Arctic sea ice volume deduced from the Pan-Arctic Ice Ocean Modeling and Assimilating
System (PIOMAS, **Zhang and Rothrock** [2003]) and the Freezing-Degree Days (FDD) based on ERA
Interim surface air temperature reanalysis. In addition we will extend the inter-comparison to the
ESA Cryosat-2 satellite measuring sea ice freeboard in order to infer sea ice thickness at the pan-
Arctic scale for the past 10 years (since the launch of Cryosat-2).
Based on the ERA Interim air temperature reanalysis at 2m altitude over the Arctic Ocean since 1979,
we calculated the number of cumulative FDD each year from September to May the following year.
From cumulative FDD we estimated sea ice growth (thickness) during the whole freezing season
based on empirical (**Anderson** [1961]) and theoretical (**Maykut** [1986]) formulations. Then from FDD



distribution both in time and space we deduced the new sea ice volume formed month after month,
year by year over the whole Arctic Ocean. We also compared Arctic sea ice volume estimates based
on FDD with PIOMAS and also with recent estimations based on ESA Cryosat-2 satellite (**Tilling et al.**
[2017]) for the freezing time period extending from September to May each year during the past 10
years. Since FDD is exclusively dedicated to the freezing season extending from September in the Fall
to May the following Spring and since Cryosat-2 sea ice thickness estimations are also limited to the
same time frame, the inter-comparison will be limited to the Arctic sea ice growth time period
starting in September-October and reaching a maximum in April-May each year.
**Arctic sea ice volume estimates over the past 40 years based on PIOMAS estimations**
In this introduction let us first concentrate on Arctic sea ice volume deduced from PIOMAS. PIOMAS
is a numerical model with components for sea ice and ocean and the capacity for assimilating data
from observations (sea ice concentration and sea surface temperature, SST). For the ice volume
simulations shown here, sea ice concentration information from NSIDC near-real time product are
assimilated into the model to improve ice thickness estimates and SST data from the NCEP/NCAR
reanalysis are assimilated in the ice-free areas. NCEP/NCAR reanalysis SST data are based on the
global daily high-resolution SST analyses using satellite and in situ observations (Reynolds et al.
2007). Atmospheric information to drive the model, specifically wind, surface air temperature and
cloud cover to compute solar and long wave radiation, are based on the NCEP/NCAR reanalysis. The
Pan-Arctic Ocean model is forced with input from a global ocean model at its open boundaries
located at 45 degrees North.  PIOMAS has been extensively calibrated and validated through
comparisons with observations from US-Navy submarines, buoys, and satellites (**Schweiger et al.**
[2011]).
A range of observations and approaches, including in situ ice thickness measurements, ICESat
retrieved ice thickness and PIOMAS model sensitivity studies, yields an uncertainty of the Arctic sea
ice volume trend of 1.0 x $10^3$ km$^3$/decade over the 1979-2010 period and a conservative estimate of
the trend over this period is -2.8 x $10^3$ km$^3$ per decade (equivalent to about 11000 km$^3$ in 39 years)
according to **Schweiger et al.** (2011). Figure 1 shows Arctic sea ice volume estimated from PIOMAS
over the past 40 years. It is important to notice that the volume decrease in winter (11000 km$^3$) is
almost similar to the drop in summer (12000 km$^3$) over the past 40 years. So the problem regarding
sea ice decline is not only related to summer melt but also to sea ice production during the freezing
time period.
An important aspect concerns the net sea ice production that is the balance between the sea ice
production during the freezing period (the black curve on Figure 1) and the sea ice ablation during
the melting season (the green curve on Figure 1).
It is interesting to compare the net ice production from year to year that we estimated (Figure 2a) by
considering sea ice growth during the freezing period extending for about 9 months (from September
to May) overlapping with sea ice melting from May to September for about 5 months. This is
equivalent to the sea ice maximum reached each year in April-May minus the sea ice maximum
reached in April-May the previous year (Figure 2a) and/or to the sea ice minimum reached in
September each year minus the sea ice minimum reached in September the previous year (Figure
2b). The main result is that even if winter sea ice production and summer sea ice melting are both
slightly increasing (black and green curves on Figure 1), the mean difference (that is the net



production) is negative most of the time for the past 40 years (5-year running mean (cyan curve) on
Figure 2a and 2b). Summing up over the past 40 years, the net sea ice loss in winter was -10703 km³
corresponding to a mean value of -274 km³ per year according to PIOMAS (Figure 2a). Equivalently
the net sea ice loss in summer was -11821 km³ with a mean value of -303 km³ per year (Figure 2b). In
addition to the long-term trend, the net sea ice volume production, Figure 2a and 2b revealed a
strong inter-annual variability, an order of magnitude higher than the long-term trend.
Interestingly, Figure 2a and 2b further indicated a 7-year oscillation characterizing the Arctic sea ice
volume internal variability. Although it is not in the scope of this paper to discuss this point, it is
relevant to mention a recent study by **Swart et al.**[2015] who looked specifically at trends in Arctic
sea ice extent for all-7 year periods between 1979 and 2013 in the observations and in 102
realizations from 31 CMIP5 models. **Swart et al**. [2015] concluded that pauses in sea ice loss such as
the one observed between 2007 and 2013 and lasting for 7 years, are fully expected to occur from
time to time. The 2007-2013 pause was following a 7-year period of intense sea ice loss from 2001 to
2007 during which **Kwok et al.** [2009] reported a total sea ice loss of 6300 km³ in four years since
2005. This huge sea ice loss occurring during the 2000s, included a massive amount of MYI attributed
to several summers characterized by no replenishment of MYI by FYI (**Kwok** [2007])
Ed Hawkins (a co-author of **Swart et al.** [2015]) suggested an analogy between Arctic sea ice behavior
and a "ball bouncing down a bumpy hill"  ([http://www.climate-lab-book.ac.uk/2015/arctic-erratic-as-](http://www.climate-lab-book.ac.uk/2015/arctic-erratic-as-expected/)
[expected/](http://www.climate-lab-book.ac.uk/2015/arctic-erratic-as-expected/)) in order to explain the combination (interaction) between "the hill" (the long-term
downward trend) and "the bumps" the internal (natural) variability  of Arctic sea ice over the past 35
years. PIOMAS estimations related to sea ice volume (rather than sea ice extent) are highlighting this
important aspect of a strong natural (internal) variability with a 7-year periodicity superimposed on a
smooth long-term trend due to increasing global temperatures of anthropogenic origin (**Gillett et al**.
[2008].
Averaged projection from 30 CMIP5 models that can better reflect the observed sea ice volume
climatology and variability indicated that the September sea ice volume minimum will decrease to
3000 km³ in the early 2060s based on a medium GHG emission scenario according to **Shu et al.**
[2015] and **Mi-Rong Song** [2016]. But this will drop to the same value (3000km³) in the early 2040s
under a high GHG emission scenario like it is today and then reach a near zero ice volume in the mid
2070s. Actually in September 2012 the Arctic sea ice volume reached an extreme low value of 3800
km³ according to PIOMAS, close to the 3000 km³ value predicted by CMIP5 models but at a much
earlier time (2012 for PIOMAS instead of 2040 for CMIP5). We will come back to future predictions
regarding Arctic sea ice in the discussion section.
A validation of the PIOMAS estimations for Arctic sea ice volume was provided by **Schweiger et al.**
(2011). It is relevant to compare PIOMAS sea ice volume estimations with other estimations such as
those deduced from cumulative FDD during the entire freezing season. The **cumulative** FDD is an old
concept similar to the ice mass budget concept used for estimating ice accumulation on glaciers and
inlandsis. The main difference comes from the snow precipitations accumulating over land for
glaciers during the entire fall-winter-spring season in contrast with the cumulative FDD in case of sea
ice over the same time period. The cumulative aspect for a long period of time (several months) in
both cases is the major factor related to new ice formation occurring during the whole freezing
season. The ice mass balance involves ablation (melting) in addition to accretion (freezing) happening



during the entire seasonal cycle. In this paper we will concentrate on the sea ice accumulation
deduced from cumulative FDD over the whole Arctic Ocean and during the entire freezing time
period from 1979 until 2018.

### 169    **2/ Methodology and data set**

We first calculated the number of freezing-degree days (FDD) based on air temperature at 2m
altitude all over the Arctic Ocean and sub-Arctic regions deduced from ERA interim reanalysis for the
period starting in 1979 until today (2018) and during the freezing time period lasting for 9 months
(September to May) each year.

### 174    **ERA Interim**

ERA-Interim is a data set based on a global climate reanalysis from 1979 to date. ERA stands for
"European Reanalysis" and refers to a series of research projects at ECMWF which produced various
datasets (including ERA-Interim, ERA-40 etc…). ERA-Interim uses a fixed version of a numerical
weather prediction (NWP) system based on a data assimilation Integrated Forecast System (IFS-
Cy31r2) to produce reanalyzed data. The system includes a 4-dimensional variational analysis (4D-
Var) with a 12-hour analysis window (**Simmons et al.** [2004] and **Berrisford et al.** [2011]).
We took advantage of the 2m air temperature produced by ERA-Interim at a spatial resolution of
0.75 degree and a temporal resolution of 6 hours from which we calculated the daily average. Then
we eliminated all the data South of 60°N and all the data on land (i.e. equal and above 0 m altitude)
using ETOPO2v2c_f4 topography. Furthermore we only considered data between September and
May the next year for the entire period extending from 1979 until 2018.

### 186    **FDD and ERA Interim**

We calculated the number of cumulative FDD for each ERA-interim grid cell over the whole Arctic
Ocean down to 60°N and during 39 years from 1979 until 2018. Cumulative FDD are progressively
increasing days after days and month by month from September to May each year.

190                       $$FDD = \int_0^t (Tf - Ta)dt \qquad \text{Eq.(1)}$$

The air temperature *Ta* at 2m altitude is provided by ERA Interim every 6 hours for each ERA interim
grid cell.   We calculated *Ta* daily average and then we estimated the cumulative FDD for each ERA
interim grid cell by integrating the difference between sea water freezing temperature *Tf* = -1.7°C
and *Ta* from September 1 to September 30 (1 month), then from September 1 to October 31 (2
months), then from September 1 to November 30 (3 months) etc…. until we covered the whole
freezing time period lasting for 9 months from September to May each year.
We also calculated the surface (km$^2$) for each individual grid cell from the ERA-Interim data file using
the following formula   [0.75*110]^2*cos[lat x]     x being the latitude.

### 199    **FDD and sea ice thickness**

Then we estimated the sea ice growth (accumulation at the bottom of sea ice) during the freezing
period extending from the Fall (September) to the following Spring (May) each year based on the
cumulative FDD over this time period based on Eq.(1).



The conductive heat flux $F_c$ throughout the ice of thickness H is  $F_c = k_i/H (T_0 - T_f)$      Eq.(2)
$T_0$ being sea ice temperature at surface and $T_f$ being sea ice temperature at bottom (i.e. sea water
freezing temperature *Tf*). $k_i$ is the sea ice thermal conductivity.
The growth rate at the base of sea ice is - $\rho_i L.dH/dt$  with $\rho_i$ the sea ice density and L the latent heat
of freezing for sea water. Considering $F_w$ being the ocean heat flux at the base of sea ice

208                    - $\rho_i L.dH/dt$  = $F_c + F_w$              Eq.(3)

$F_c$ is in equilibrium with $F_a = C_a (T_a - T_0)$ the heat exchange between the ice and the atmosphere at
surface, $C_a$  being an average transfer coefficient taking into account the sensible, latent and net long
wave heat exchange but neglecting the solar short wave radiation (negligible during the polar night).
Solving for $T_0$ and assuming the ocean heat flux at the base of the ice $F_w << F_c$ the integration over
time with H = 0 at time t = 0 will give  $H^2 + 2k_i H/C_a = 2 k_i/\rho_i L$ . FDD        Eq.(4)
Including a snow layer of thickness $h_s$ leads to the relationship suggested by **Maykut** [1986]

215                    $H^2 + (13.1 h_s + 16.8) H = 12.9$ FDD          Eq.(5)

Equation (5) was also used by **Harpaintner et al.** [2001] for estimating ice production in Storfjorden,
Svalbard.

## 3/ Results

### FDD and Sea Ice thickness

Figure 4a shows monthly cumulative FDD spatial distribution as a function of FDD for the period
extending from September 2016 until May 2017 and the same on Figure 4b for the period extending
from September 2017 until May 2018.
The comparison between the 2 periods is highlighting the strong inter-annual variability
characterizing Arctic sea ice. Another important aspect concerns a sharp double peak FDD spatial
distribution that we had not anticipated initially (Figure 4a). We will see in the following that the
double peak-like FDD distribution is reflecting very closely the well-known sea ice thickness
distribution typical of FYI and MYI in the Arctic Ocean. 2017 (Figure 4a) appears like a very abnormal
year characterized by the lowest cumulative FDD over the past 40 years (Figure 5). 2018 also appears
like a very abnormal year characterized by a single peak FDD spatial distribution (Figure 4b). The
second peak vanished in 2018 and as we will see later on, it corresponds to the MYI extinction all
over the Arctic Ocean.
Figure 5 illustrates the drastic 9 month cumulative FDD decrease when comparing the 1980s and the
1990s with more recent years (2000s and 2010s). The 9 month cumulative FDD reduction during the
past 40 years amounted to about 2000 FDD at the peak values extending over more than $1.2x 10^6$
$km^2$ in the Arctic Ocean.



As shown on Figure 6, the coldest region characterized by the highest 9 month cumulative FDD
values, is clearly located north of the Canadian Archipelago and Greenland and corresponding to the
Western Arctic North of the American Continent.

In contrast the Eastern Arctic (North of Eurasia) is the region experiencing the warmest temperature
increase characterized by the strongest reduction of cumulative FDD distribution over the past 40
years and in particular during the most recent 10 to 20 years (Figure 7). During the past 10 years the
warming of the entire Eastern Arctic is spectacular, which explains the new and strong interest for
exploiting the Northern Sea Route for shipping activities (**Gascard et al.** [2017]) where sea ice
conditions were less severe than during the last part of the 20th century.
Figure 8 represents Sea Ice thickness distribution  for the period 2010 until 2018 deduced from
cumulative FDD and based on the **Maykut** [1986] formulation (Equation 5) for $h_s$ = 0. There are
important and interesting aspects we would like to comment on Figure 8.
1/A double peak sea ice thickness distribution typical of the Arctic Ocean and related to FYI and MYI
is clearly visible on Figure 8 except for 2018 when the second peak corresponding to the MYI,
disappeared. During the past decades, it was quite often reported in several publications (**Kwok et al.**
[2009] that MYI was vanishing but 2018 appeared as a very abnormal year characterized by a
complete MYI extinction for the first time ever over the past 40 years according to FDD sea ice
thickness distribution.
2/ 2017 looked like quite an abnormal year too, characterized by a significant FYI thinning of about
15cms at the peak compared to the previous (and to the following) years. The first peak
corresponding to FYI (2.04m thick ice in 2017 and 2.19m thick in 2018) extending over 1.2 x $10^6$ km$^2$
approximately, is quite sharp for both years. **Stroeve et al.** [2018] described a broad region of
anomalously thin ice in April 2017 relative to the 2011-2017 mean thickness values. Based on Los
Alamos sea ice model simulations (CICE), **Stroeve et al.** [2018] estimated a thinning of about 11 to 13
cm in 2017, very similar to the 15cm based on cumulative FDD sea ice thickness for the same time
period.
Figure 9 illustrates the sea ice thinning over the whole Arctic Ocean during the past 40 years and in
particular within the Arctic peripheral shallow seas (Chukchi, East Siberian, Laptev and Kara Seas) but
also within the central Arctic Ocean and the Beaufort Sea. The thicker ice is still located North of
Greenland and the Canadian Archipelago, and the thinner ice is located North of Eurasia. MYI
disappeared almost entirely except for the few remnants located in the Lincoln Sea.
**FDD and sea ice volume**
Having estimated sea ice thickness distribution in space and time, we can now estimate Arctic sea ice
volume month by month from September to May each year starting in September 1979 and ending
in May 2018. The results are presented on Figure 10a.
We have estimated the error for sea ice volume based on FDD and attributed to an initial error of
0.6°C in the ERA-Interim 2m air temperature data file. Considering a 1% error in sea ice extent (i.e.
about 100 000 km$^2$) we came about a 4% error in sea ice volume (equivalent to 1000 km$^3$). This is



quite comparable to the Arctic sea ice volume error estimated by **Schweiger et al.** [2011] for
PIOMAS.
It is interesting to note that both 2017 and 2018 are the years characterized by the lowest sea ice
winter production when compared to the previous 38 years. Based on FDD estimations the sea ice
volume decreased by about 5000 km$^3$ in 40 years (Figure 10a) and this is about half of the PIOMAS
sea ice decrease estimations (Figure 10b). The difference is mainly explained by the fact that sea ice
thickness based on a cumulative FDD is assuming H = 0 at t = 0. Consequently FDD cannot explicitly
account for MYI. The fit between FDD and PIOMAS improved in recent years due to a quasi
disappearance of MYI.
It is remarkable to note (Figure 10a and 10b) that in February 2018 the Arctic sea ice volumes
estimated both from PIOMAS and FDD were exactly the same (18500 km$^3$). Another interesting
comparison between FDD and PIOMAS sea ice volume estimates concerns the maximum sea ice
volume value that is reached in April for PIOMAS and in May for FDD. This is quite logical since in
addition to the freezing still active in May as indicated by FDD, PIOMAS is also taking care of the sea
ice melting that already started in May and overlapping with some active freezing. FDD is only
accounting for the freezing and not for the sea ice melting. In another paper we will introduce
melting-degree days (MDD) overlapping FDD during the Fall and Spring seasons.
It should also be mentioned that a thin layer of snow on top of sea ice is reducing the difference
between FDD and PIOMAS sea ice volume estimates as shown on Figure 11. A 5cm snow layer is
increasing very significantly the importance of the linear term in Equation (5) and is producing less
sea ice volumes quite comparable to those  obtained from the linear relationship previously
introduced (Figure 3) and shown on Figure 11. It is remarkable to note the excellent fit between the
PIOMAS and the FDD linear relationship for estimating sea ice volume for recent years involving not
only the long-term trend but also the strong inter-annual variability component. On Figure 11 we
also represented an FDD based relationship including a 10cm deep snow layer on top of sea ice. In
that case, the sea ice volume, very sensitive to the snow layer depth, was leading to an excessive
reduction of the sea ice volume. Regarding the Anderson's experimental relationship introduced on
Figure 3, it was based on a limited time period (1 month) for cumulative FDD limited to a maximum
range of 2000 FDD. Clearly the Anderson's experimental relationship is not appropriate for a much
longer time frame as the one we considered in this paper (up to 9 months) and corresponding to
much higher cumulative FDD values (6000 FDD range).
The so called "snow ice" process was recently observed in the Arctic Ocean.  It resulted from a
refrozen thick snow layer flooded by sea water on top of sea ice. Due to an excessive snow load on
top of ice floes, a thick snow layer following abundant snow precipitation in a wetter Arctic is
reducing the freeboard of ice floes to zero and/or even to negative values. This process is only
affecting certain regions of the Arctic but not the whole Arctic Ocean. In this paper we are
considering the large scale effect of a thin snow layer on top of sea ice involving a positive freeboard
and no "snow ice" effect.
**PIOMAS, FDD and Cryosat-2 sea ice volume intercomparison**
Finally we compared PIOMAS and FDD sea ice volume estimates together with Cryosat-2 (Figure 12)
based on results published by **Tilling et al.** [2017]. In fact FDD based sea ice volume maximum are



surprisingly similar to Cryosat-2 estimates and at the same time very consistent with PIOMAS
estimates as previously described. However we observed a few differences. As mentioned before,
FDD sea ice volume starts from 0 each year since ice thickness H = 0 at t = 0 in contrast with Cryosat-
2 and PIOMAS taking into account the old ice (MYI) remaining at the end of each summer season (or
early fall) in the Arctic Ocean.  But due to MYI's recent collapse, Arctic sea ice volume estimations
based on FDD, PIOMAS and Cryosat-2 are now converging. We previously mentioned that the sea ice
volume maximum is reached in May for FDD based sea ice volume estimations compared to April for
both PIOMAS and Cryosat-2 due to some sea ice melting overlapping with sea ice freezing still active
in May. But overall the three approaches are giving remarkably similar results. As already mentioned,
due to melt ponds on the ice in the melt season which confuses the sea ice thickness retrieval,
Cryosat-2 is not able to reliably estimate sea ice thickness (and volume) during the melting season.

## 4/ Discussion

First of all, it should be clearly stated that our choices for ERA interim (surface air temperature) and
PIOMAS (Arctic sea ice volume estimations) are purely arbitrary. There are other potential choices of
course and a useful and interesting additional work would be to evaluate and to inter-compare
several other options since they are all affected by some intrinsic biases and errors (**Jakobson et al.**
**[2012]**). For simplicity we did not want to open a new section by inter-comparing different data set
and co-evaluating various models. We cited references providing indications about the relevance of
the data set (ERA Interim) we chose and the numerical model (PIOMAS) we selected. The novelty of
this paper is the application of the old FDD concept to an up to date data set (ERA Interim) and an
inter-comparison with a well known numerical model (PIOMAS) and modern technologies (altimeter
on board the ESA Cryosat-2 satellite). All the numbers presented in this work are considered relevant
in a relative rather than absolute sense.
In this paper we applied the cumulative FDD concept to ERA interim air temperature at 2m altitude
above sea ice, in order to evaluate Arctic sea ice volume formed during the freezing season at a pan-
Arctic scale and over the past 40 years. The important point concerns the **cumulative** aspect applied
to FDD calculations over a long period of time (up to 9 months). The cumulative concept is also used
for estimating ice accretion on top of glaciers or ice caps located on a continent based on snow
accumulation over the Fall-Winter-Spring season at surface. In case of sea ice, the cumulative FDD
are responsible for sea ice formed at the base of the ice instead of snow accumulation on top of a
glacier for ice over land. In contrast it is well recognized that the snow accumulating on top of sea ice
is slowing down the formation of ice at the base of sea ice rather than contributing to ice growth in
case of glaciers and inlandsis.
The Arctic sea ice thickness distribution in space and time directly inferred from the FDD reveals the
typical double peak distribution characterizing the Arctic FYI versus MYI disposition. However, to our
biggest surprise, a single peak sea ice thickness distribution appeared in 2018 for the first time ever
over the past 40 years. The second peak disappeared in 2018 and this can be interpreted as a
manifestation of a quasi extinction of the Arctic MYI in 2018. According to PIOMAS, the Arctic sea ice
loss in winter amounted to more than 10000 km$^3$ over the past 40 years (Figure 1 and Figure 10b).
This is twice as large as the sea ice loss deduced from FDD (Figure 10a) over the same time period.
The difference is mainly due to the Arctic MYI not explicitly included in FDD estimations. Since Arctic
MYI vanished recently, the sea ice volume deduced from PIOMAS and FDD are now remarkably



similar. In February 2018 the sea ice volume deduced from FDD and PIOMAS were identical (18500
km$^3$) and the lowest over the past 40 years. It is now dangerously approaching the amount of sea ice
volume melting every year. For the past 10 years, it appeared like the PIOMAS Arctic sea ice volume
decayed at a lower rate than during the previous 10 years (2000-2010) and followed closely the
Arctic sea ice volume maximum deduced from FDD estimations either based on a linear relationship
relating FDD and sea ice thickness or a quadratic relationship with a 5cm layer of snow on top of sea
ice. Not only is the trend similar but also the inter-annual variability between the FDD and PIOMAS
sea ice volume estimations is remarkably similar.
Almost every year since the late 70s, we were able to identify an "abnormal" situation in the Arctic
characterized by some extreme conditions related to sea ice extent, thickness and/or volume. The
most recent years (2017-2018) are no exception. Both years were also characterized by remarkable
"anomalies" in particular concerning Arctic sea ice volume reaching an extreme minimum in 2017
closely followed in 2018 compared to the previous 40 years for both FDD and PIOMAS sea ice volume
estimates.  **Perovich et al.** [2017] mentioned "the lowest winter maximum ice extent in the satellite
record (1979-2017) which occurred on 7 March 2017. The extent was 14.42 million km$^2$, 8% below
the 1981-2010 average". In March 2017 Arctic sea ice volume estimations were 15% less than the
1981-2010 average according to FDD-based estimations (Figure 10a) and 20% less according to
PIOMAS (Figure 10b).
The strong inter-annual variability superimposed to the long-term trend is remarkably similar
between cumulative FDD-based sea ice volume estimates and PIOMAS in particular for the most
recent years. Pan-Arctic sea ice volume inferred from cumulative FDD is quite comparable to Arctic
sea ice volume estimated from PIOMAS and also from Cryosat-2 for the past 7 years when the
amount of MYI was particularly low. A remarkable 7-year oscillation appeared in the 40 years
PIOMAS sea ice volume time series. According to **Swart et al.** [2015], this oscillation could be
interpreted as an analogy to a "ball bouncing down a bumpy hill." The "hill" would represent the
long-term decline of Arctic sea ice and the "bumps" the natural internal variability of Arctic sea ice.
This 7-year oscillation still needs further investigation.
It is clear that the long-term Arctic sea ice volume decrease revealed by PIOMAS, is due to an
asymmetry between sea ice formation (accretion) during the freezing period and sea ice ablation
during the melting period. Over the past 40 years, the annual mean net ice production was negative
(-300 km$^3$ per year), accounting for a total loss of more than 10000 km$^3$ of Arctic sea ice for the past
40 years. According to FDD based sea ice volume estimations, half of the total sea ice loss concerns
FYI and the other half concerns MYI according to PIOMAS.
An asymmetry was also identified by **Bathiany et al.** [2016] "On the potential for abrupt Arctic winter
sea ice loss". Estimating sea ice formation during the freezing season starting in September and
lasting for about 9 months until April-May the following year, is as important as estimating Arctic sea
ice melting from May to September every year. The long-term decrease of Arctic sea ice is not only
due to a warmer and longer summer season but also to a milder and shorter freezing time period.
The new sea ice formed every year during the freezing time period has slightly increased, but not as
much as the sea ice that has melted away during the melting season. This is compatible with milder
(warmer) winters producing more FYI mainly due to more open water and thinner ice after a steeper
ice decrease in summer.  Figure 10a indicates the sea ice loss was much larger during the winter



months than during the fall. In other words the overall decrease of the Arctic surface air temperature
during the freezing period is the dominant factor to explain the Arctic sea ice loss during winter
rather than a shortening of the freezing period. Winter temperature rise is the dominant factor
compared to winter time reduction for attributing the main part of the Arctic sea ice losses.
During the past 3 winters, meteorologists reported about major disturbances of the weather
patterns occurring in the northern hemisphere such as atmospheric blocking over Scandinavia, cold
air outbreaks over western Europe and North America, subtropical warm air invasion up to the North
Pole by cyclones, split polar vortex and jet stream instabilities all related to Arctic sea ice drastic
changes and extreme Arctic temperatures (**Overland and Wang** [2010], **Tang et al.** [2013], **Moore et**
**al.** [2018]. It seems quite clear that warm air masses carried on by extra-tropical cyclones (**Moore**
[2016], **Kim et al.** [2017]) have an increased tendency to invade the whole Arctic Ocean in winter
time strongly impacting Arctic sea ice everywhere from Eurasia to North America and from the Fram
Strait on the Atlantic side to the Bering Strait on the Pacific side.
The cumulative FDD distribution in space and time revealed an asymmetry between the Western
Arctic (North of Greenland and Canada) being the coldest Arctic region and the Eastern Arctic (North
of Eurasia) experiencing the strongest warming. This will have important consequences for human
activities such as shipping and navigation in polar regions (**Gascard et al.** [2017]). We can seriously
predict a quasi disappearance of the Arctic sea ice in summer during the coming decades.
Undoubtedly this is a major event that will deeply affect marine and terrestrial ecosystems
(Lawrence et al. 2008), weather conditions in the northern hemisphere (**Overland and Wang** [2010])
and strongly impacting human activities globally (**Crepin et al**. [2017]).
In light of our findings regarding cumulative FDD and related Arctic sea ice volume estimates mainly
based on ERA interim surface air temperature, it is relevant to mention a study by **Pithan and**
**Mauritsen** [2014] notably Arctic amplification dominated by temperature feedbacks in contemporary
climate models. Albedo feedbacks (often cited as the main contributor by **Serreze and Francis** [2006]
and many others) are of a secondary importance during the freezing season for which we neglected
the short wave incoming radiation. This is another justification for applying the FDD concept to
estimate Arctic sea ice volume in winter not directly involving the albedo effect.
Over the past 40 years, Arctic sea ice volume has been reduced by 75% in summer. Today the Arctic
sea ice volume maximum in April-May (about 22000km$^3$) is getting close to the Arctic sea ice volume
minimum in September (16000 km$^3$), 40 years ago. Based on a steady loss of -300km$^3$ of sea ice per
year and a Pan Arctic sea ice volume minimum estimated to be about 4000 km$^3$ at the end of the
summer by now, it should take no more than 12 to 15 years to melt away the remaining 25% of sea
ice still resisting the summer melt. Accordingly a blue Arctic should appear in summer 2030-2035
much sooner than predicted from CMIP5. The IPCC AR5 concluded that it is likely the Arctic would be
reliably ice-free in September by 2050 assuming high future GHG emission scenarios. Here "reliably
ice free" meant five consecutive years with less than $10^6$ km$^2$ of sea ice extent. The expected
outcome is that the long-term decline in Arctic sea ice extent, thickness and volume will continue as
global temperatures increase. There will be a further "ball bouncing down the hill" effect (both up
and down) and consequently there will be few years becoming ice-free in summer during the 2020s,
2030s or 2040s depending on future GHG emissions impacting on the Arctic sea ice long-term trend
and on the natural (inter-annual) variability as well.



## 5/ Summary

During the past 40 years, the drastic reduction of Arctic sea ice captured scientific headlines, media attention and large public audience as the most glaring evidence of climate change on planet Earth. Concerning Arctic sea ice, it is worthwhile to note that the first parameter identified as an indicator of climate change concerned sea ice thickness resulting from US submarines upward looking sonar measurements during the 1990s. This was followed by Arctic sea ice extent in the 2000s highlighted by the 4th IPY (2007-2008) and the spectacular sea ice extent reduction occurring during the 2007 summer. This led to the so-called Sea Ice (extent) Outlook initiative largely based on satellite observations and numerical models for short-term Arctic sea ice prediction. Arctic sea ice volume is a more recent challenge mainly due to the difficulty in measuring and/or estimating sea ice volume with a decent accuracy (i.e. $+/- 1000 km^3$). Arctic sea ice volume evolution over the past 40 years was characterized by a long-term trend superimposed on a strong inter-annual variability highlighting a 7-year oscillation that still needs to be analyzed in order to identify its origin and its cause. Since IPY, 10 years ago, the on-going processes affecting/impacting Arctic sea ice have continued, amplified and accelerated. In 2018 we were witnessing for the first time a quasi disappearance of the Arctic Multi Year Ice (MYI).

Here are some of the main outcomes resulting from this study:

1/ Arctic sea ice volume is a main parameter related to climate change. It is more sensitive than Arctic sea ice extent or Arctic sea ice thickness taken separately. Compared to the situation 40 years ago, the Arctic sea ice volume minimum has decreased by 75% in summer (from about 16000 $km^3$ down to 4000 $km^3$) compared to a 50% decrease for both the Arctic sea ice extent and Arctic sea ice thickness. The absolute Arctic sea ice minimum was reached in September 2012 (3.4 $10^6$ $km^2$ in extent and 3800 $km^3$ in volume).

2/ Based on PIOMAS sea ice volume estimations, we confirmed the prediction of **Wang and Overland** [2009] for a blue (summer ice-free) Arctic Ocean by 2037 or even earlier (2030-2035). Considering that today 75% of the Arctic sea ice volume melted during summer, it would not take long to melt away the remaining 25% of Arctic sea ice in summer.

3/ IPCC models dealing with Arctic sea ice have significantly improved from AR4 to AR5 IPCC reports but they are still lagging behind reality by 10 to 20 years based on Arctic sea ice volume best estimations.

4/ Due to a strong Arctic sea ice volume natural inter-annual variability superimposed with a smoother long-term trend of anthropogenic origin (which is an order of magnitude smaller than the inter-annual variability), it is likely there would be an ice free Arctic Ocean in summer one year or another during the 2020s, 2030s and 2040s.

5/ A 7-year oscillation appeared clearly in the net ice production estimated from PIOMAS. This oscillation could very well be the expression of a natural internal variability in response to a global warming of anthropogenic origin. A precise attribution to the origin and cause of this oscillation would improve Arctic sea ice prediction.

6/ The sharp double peak spatial distribution of cumulative FDD over time is also indicative of a significant contribution from Ocean fluxes at the bottom of the Arctic sea ice (**Fan et al.** [2017]).



Ocean fluxes should be included in the Arctic sea ice budget as well as cumulative Melting-Degree
Days (MDD) to complete the sea ice seasonal cycle. Both aspects will be the topic of another
publication.
7/ The cumulative FDD sea ice thickness-based estimations revealed a quasi disappearance of the
MYI for 2018. This is the first time ever this remarkable event did occur over the past 40 years. That
also explains the reason why PIOMAS sea ice volume estimations are fitting much better FDD sea ice
volume estimations for recent years since FDD can only take into account the new ice (FYI) formed
each year. Arctic sea ice volume maximum differences of about 5000 km$^3$ between PIOMAS and FDD
based estimations in the past (1980s and 1990s) were mainly due to the abundant Arctic MYI that
has vanished by now.
8/ A thin snow layer (few centimeters) on top of sea ice is a very sensitive parameter to better
estimate the contribution from cumulative FDD for sea ice formation and sea ice volume estimations.
9/ The similarities between sea ice volume based on FDD and PIOMAS confirms the temperature
feedback is a primary contributor to the Arctic sea ice growth in winter rather than the albedo
feedback more efficient during the summer season.
10/ There is a large asymmetry between winter freezing and summer melting in the Arctic but also
between the Western Arctic and the Eastern Arctic. The Western Arctic is significantly colder than
the Eastern Arctic with the later is experiencing the strongest warming. That will have important
impacts on the development of human activities such as polar shipping and resource extraction in
the Arctic (**Crepin et al.** [2017]).
11/ The primary importance of surface air temperature is highlighted by FDD sea ice volume
estimations for the Arctic Ocean. This is also supporting recent studies led by meteorologists
(**Overland et al.** [2016], **Binder et al.** [2017] regarding increasing Arctic air temperatures related to
large scale atmospheric circulation (atmospheric blocking, cold air outbreaks, split polar vortex,
cyclonic activity) strongly impacting mid latitude weather systems (snow precipitations, floods and
drought etc..) in the Northern hemisphere (**Cullather et al.** [2016]).
12/ The three most recent winters 2015-2016, 2016-2017 and 2017-2018 produced the smallest
amount of sea ice over the past 40-year winter time period according to both FDD and PIOMAS Arctic
sea ice volume estimations.
One of the main objectives of this paper dealt with an Arctic sea ice volume inter-comparison
involving PIOMAS, FDD and Cryosat-2 in order to make progress towards more accurate and reliable
Arctic sea ice characteristics estimations and predictions. Sea ice volume is a challenging and
extremely important element of the Earth's climate system due to its greater sensitivity to climate
change compared to sea ice extent and sea ice thickness taken separately. Accordingly sea ice
volume deserves much more attention for future Arctic studies. We would like to suggest using more
extensively Arctic sea ice volume deduced from cumulative FDD in particular to evaluate the impact
of climate change on Planet Earth in the future.
**Acknowledgments**

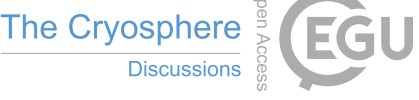

We would like to thank the EU projects ACCESS (EU Grant Agreement N°265863) and ICE-ARC (EU
Grant Agreement N°603887) and the Pan Arctic Options Belmont Forum project (ANR-14-AORS-003-
01). ERA Interim data set can be found at
ftp://ftp.climserv.ipsl.polytechnique.fr/erai/GLOBAL_075/4xdaily/AN_SF/
JZ is supported by the NASA Cryosphere program (NNX17AD27G). The PIOMAS sea ice volume data
we used are available at http://psc.apl.uw.edu/research/projects/arctic-sea-ice-volume-anomaly/.
Regarding sea ice thickness PIOMAS data are available at
http://psc.apl.washington.edu/zhang/IDAO/data_piomas.html.
We also would like to thank R. Tilling and A. Ridout (UK Centre for Polar Observation and Modelling)
for allowing us to make use of the Cryosat-2 data available at
http://www.cpom.ucl.ac.uk/csopr/seaice.html.

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



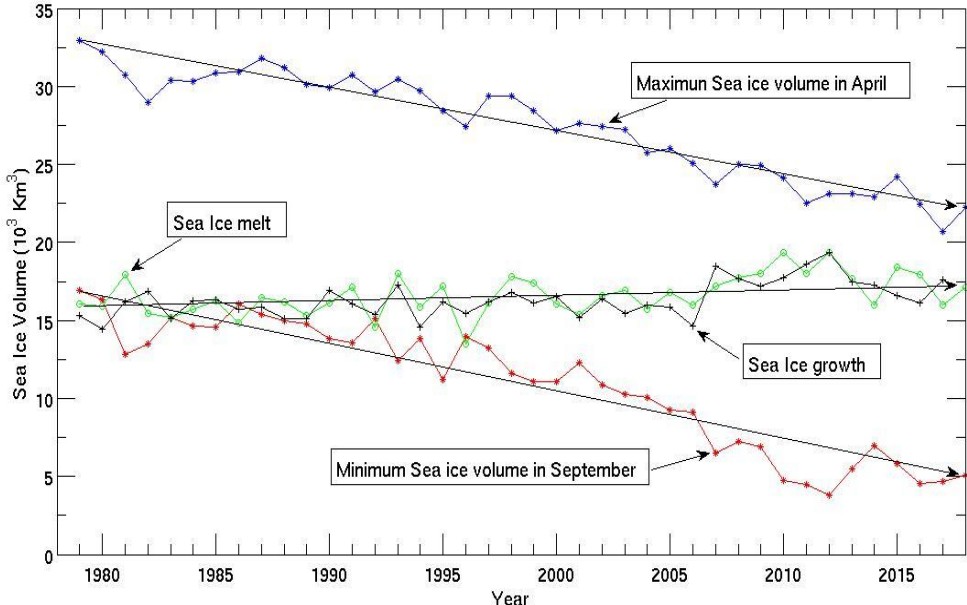


Figure 1. Arctic sea ice volume maximum in April (blue curve) and sea ice volume minimum in
September (red curve) each year according to PIOMAS from 1979 to 2018. The green curve
represents the sea ice volume melting from the maximum in April to the minimum in September
each year and the black curve represents the sea ice volume formed each year from the minimum in
September to the maximum in April the next year.




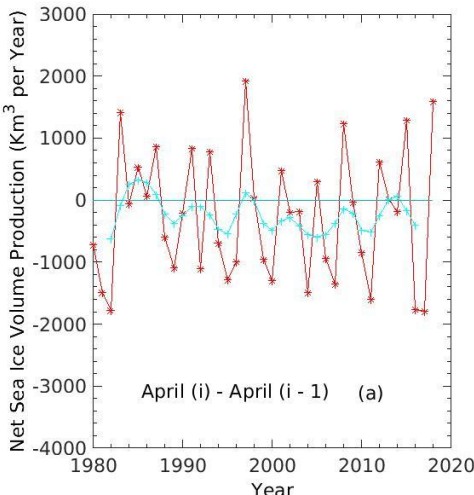 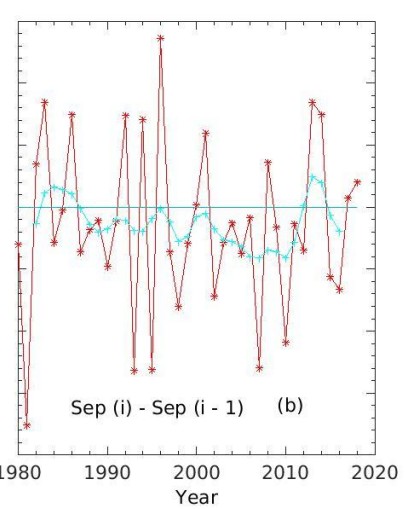


Figure 2a. Net Arctic sea ice volume production in April from 1979 to 2018 according to PIOMAS.
Figure 2b. Net Arctic sea ice volume production in September from 1979 to 2018 according to
PIOMAS. The cyan curve represents the 5 year running mean.

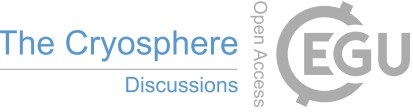



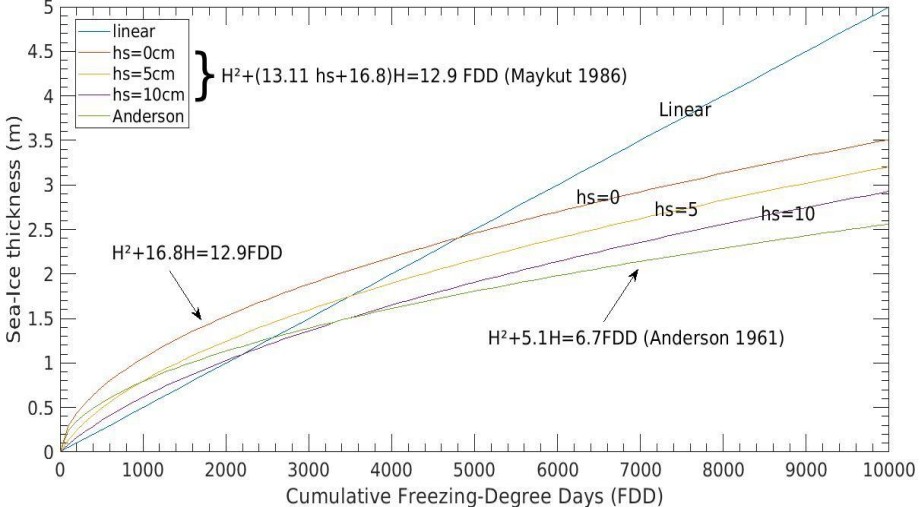


Figure 3. Sea ice thickness (H in meters) as a function of cumulative FDD based on 1/ a linear
relationship, 2/ a theoretical relationship (Equation 5) proposed by Maykut (1986) considering a 0cm,
5cm and 10cm snow layer thickness $h_s$ on top of sea ice and 3/ an experimental formulation
proposed by Anderson (1961).

656



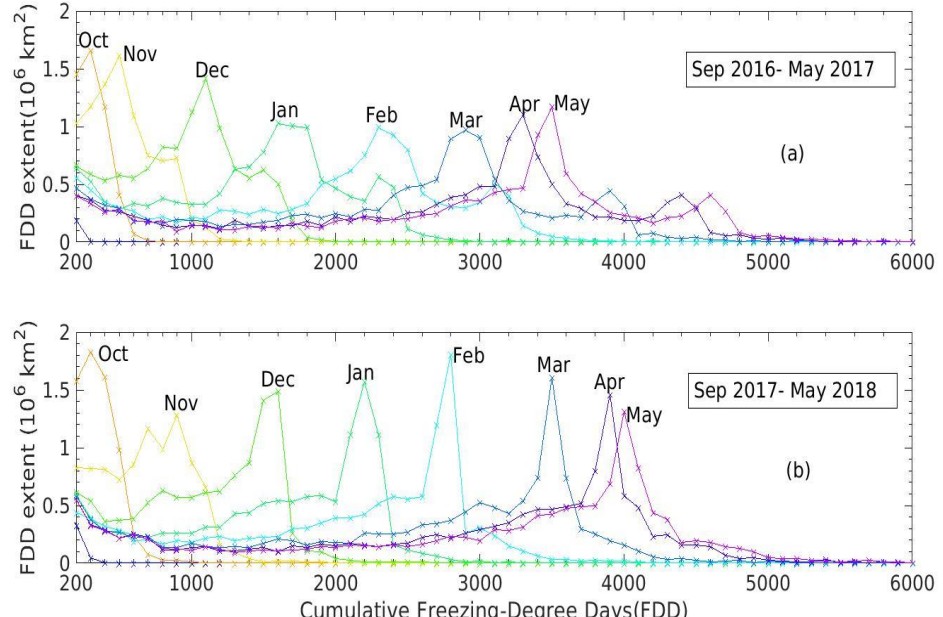

657

Figure 4a. Spatial distribution (extent) of cumulative FDD for 9 time range periods covering the
freezing season starting in September 2016 for the time range period 1 (30 days) and ending in May
2017 for the time range period 9 (270 days). Figure 4b. The same as figure 4a but for the time period
starting in September 2017 until May 2018.




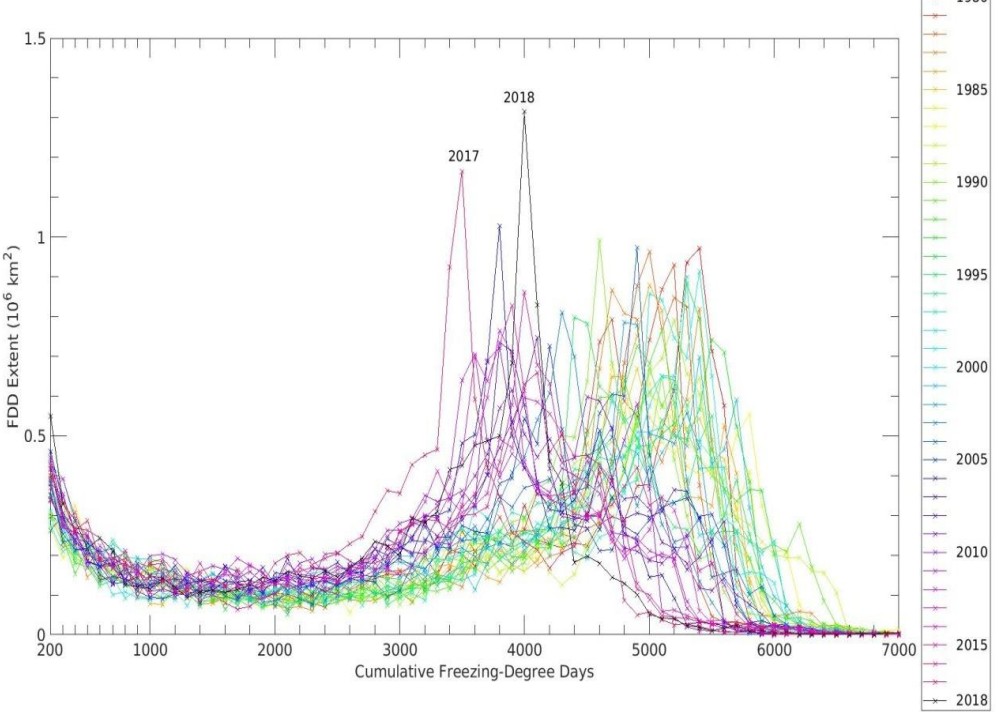


Figure 5. Spatial distribution (extent) in May of 9 month cumulative FDD (September until May) from
1980 until 2018 each year.



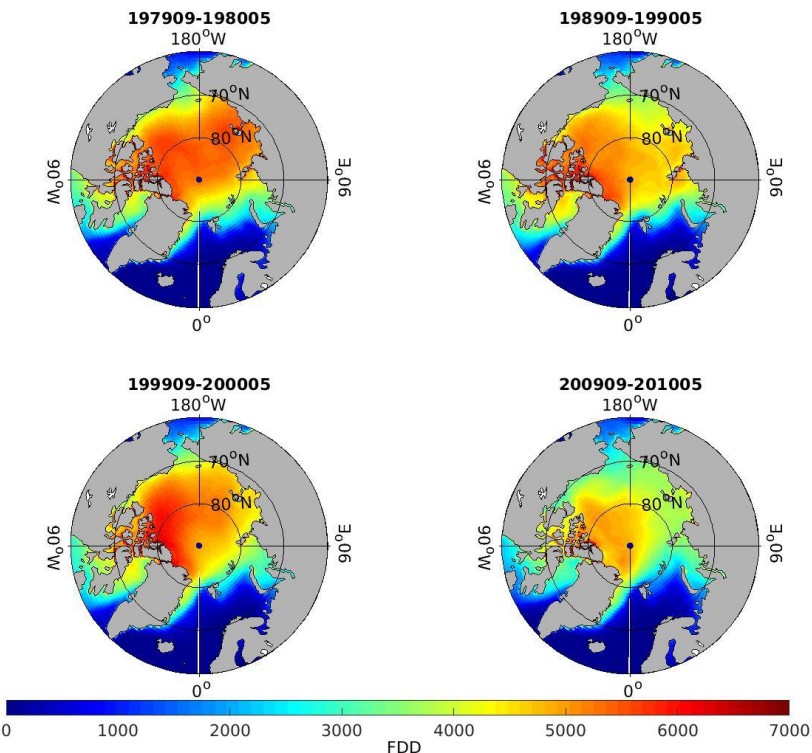

Figure 6. Maps representing the spatial distribution in May of 9 month cumulative FDD (September to May) for 4 different years (1980, 1990, 2000, 2010) over the whole Arctic Ocean.





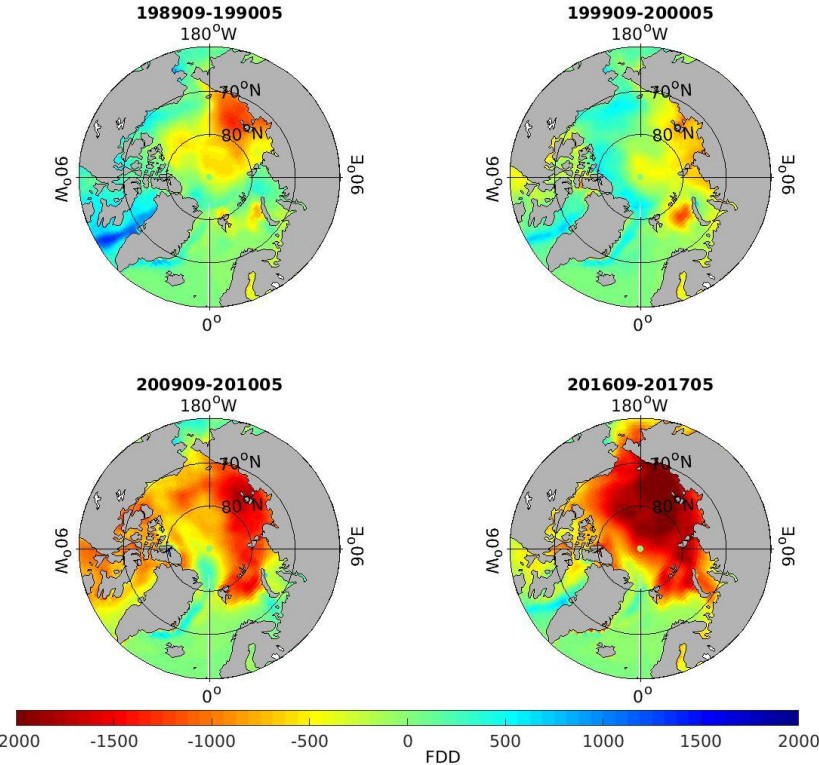


Figure 7. Maps representing the spatial distribution in May of cumulative FDD differences for a 9
month time range period (September to May) relative to 1980 used as a reference for 4 different
years (1990, 2000, 2010, 2017) over the whole Arctic Ocean.


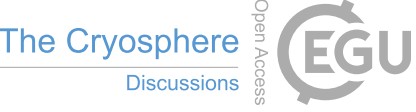




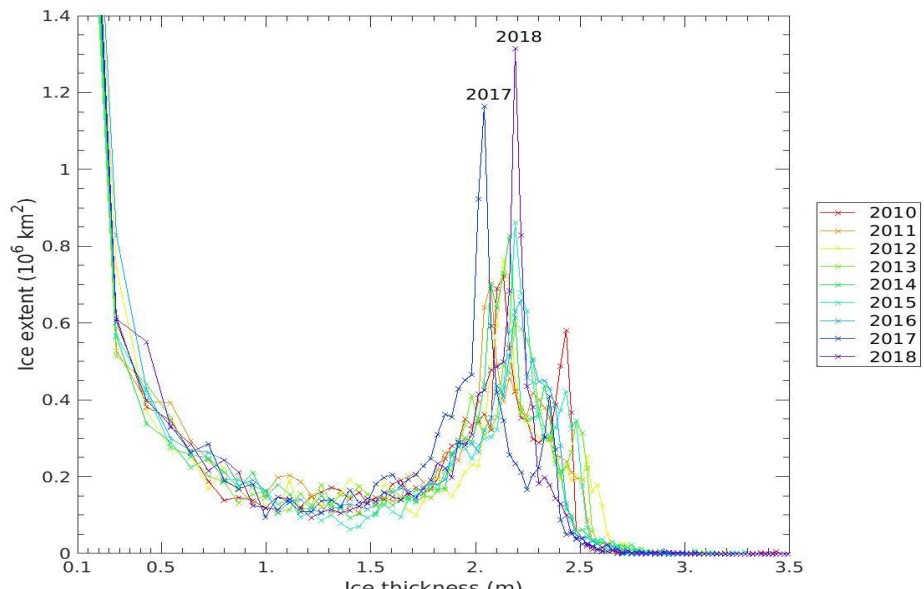

Figure 8. Sea ice thickness distribution in May deduced from 9 month cumulative FDD time range
period (September to May) from 2010 until 2018 and based on Equation 5 for a snow layer $h_s = 0$.




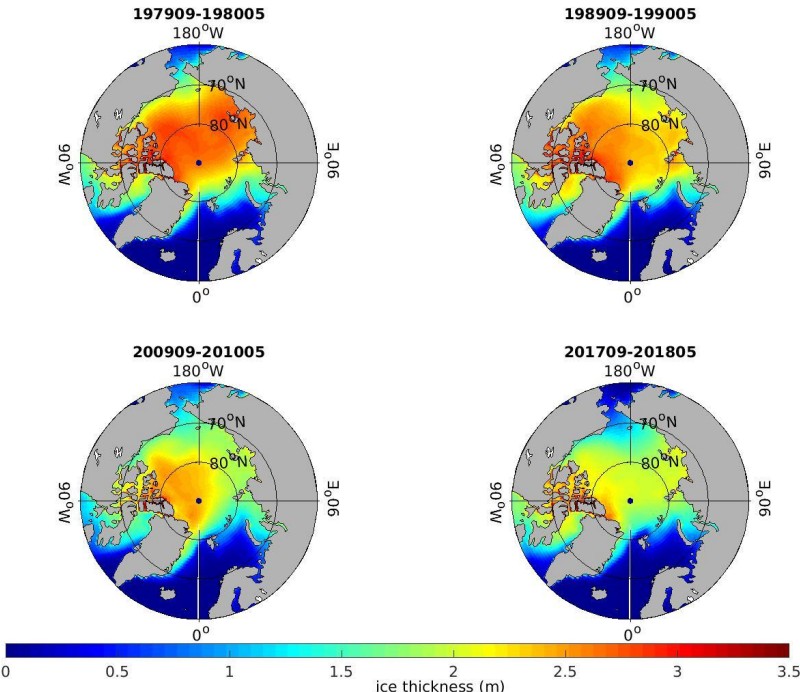


Figure 9. Maps representing Arctic sea ice thickness (m) spatial distribution in May based on 9 month
cumulative FDD time range period (September to May) and Equation 5 ($h_s$ = 0) for 4 different years
(1980, 1990, 2010 and 2018).




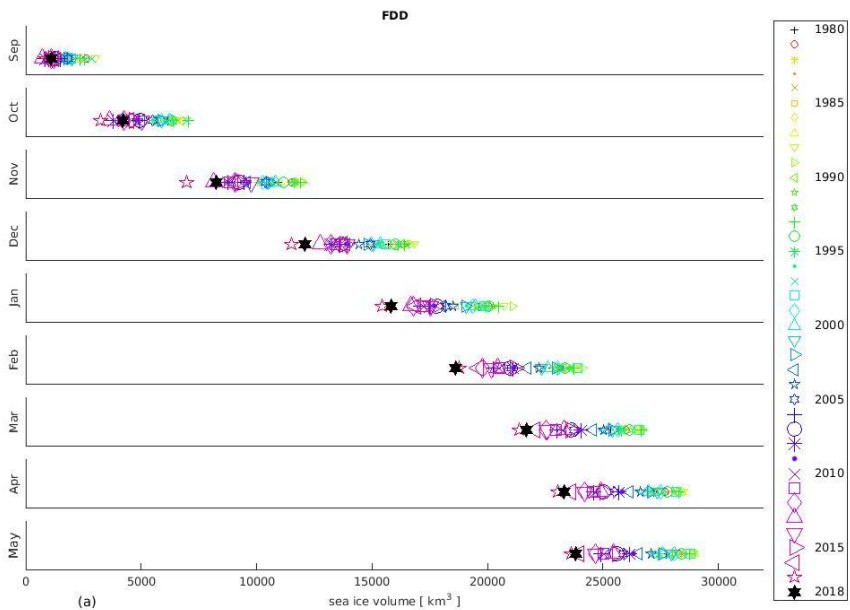


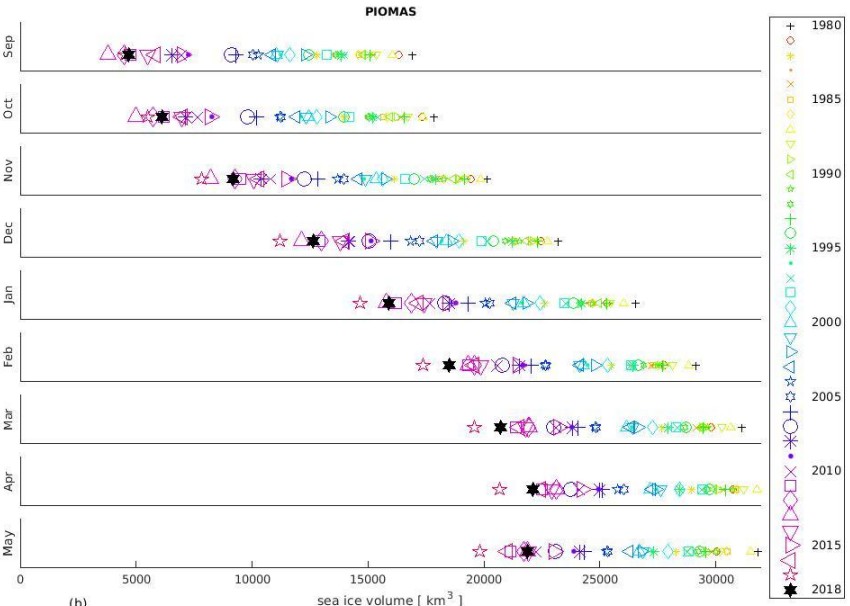


Figure 10a. Arctic sea ice volume maximum each month deduced from cumulative FDD and Equation
(5) for $h_s = 0$ during the freezing season from September to May each year from 1980 until 2018.
Figure 10b. Same as figure 10a according to PIOMAS.



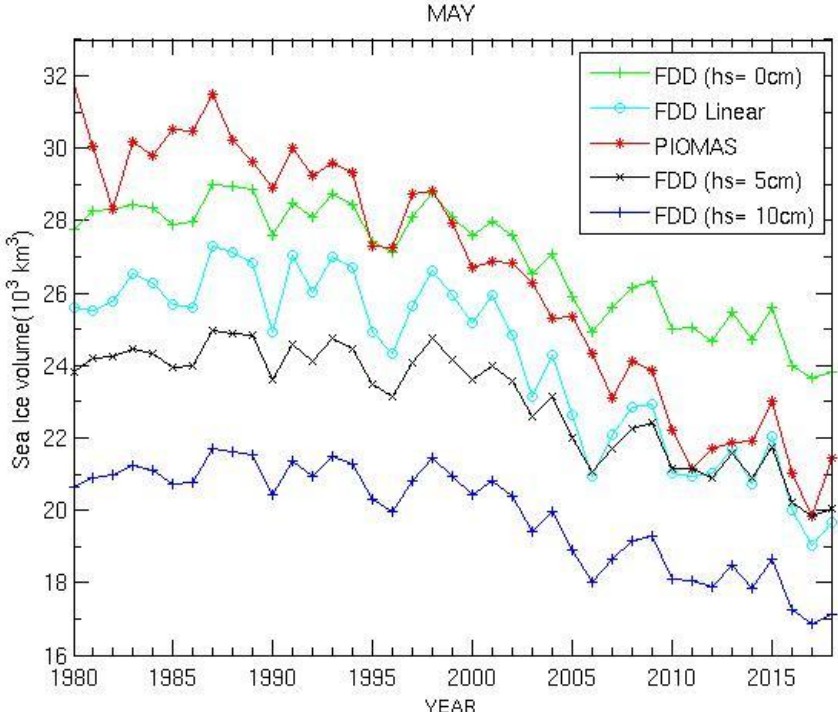


Figure 11. Arctic sea ice volume maximum in May each year from 1980 until 2018 deduced from a/
PIOMAS (red curve),  b/ the linear relationship (figure 3) relating a 9-month cumulative FDD and sea
ice thickness (cyan curve), c/ the Maykut's relationship (equation 5 and Fig.3) relating a 9 month-
cumulative FDD and sea ice thickness for $h_s$ = 0 (green curve), for $h_s$ = 5 cm (black curve) and for $h_s$ =
10 cm (blue curve). $h_s$ is the snow layer thickness on top of sea ice.








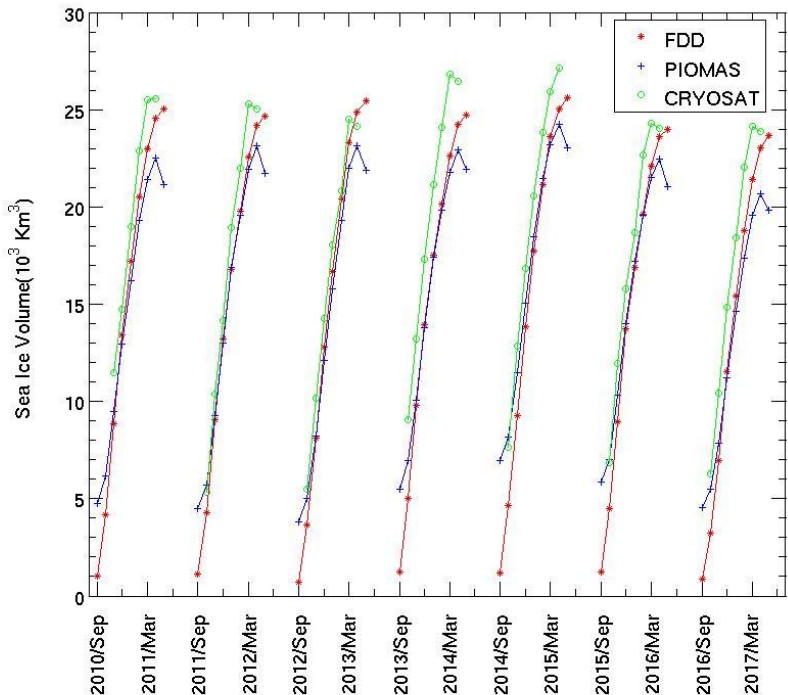

Figure 12. Arctic sea ice volume inter-comparison between FDD (red), PIOMAS (blue) and CRYOSAT
(green) for the period 2010 until 2017. Sea ice volumes deduced from 9 month cumulative FDD are
based on Equation 5  for $h_s = 0$ (Figure 3).