# Peer review of "Rapid decline of Arctic sea ice volume: Causes and consequences"

_The Cryosphere, 2019_

## Short Comment (SC1) · 16 Jan 2019

At an average loss of 300km^3/year the arctic is removing 1e20 joules of heat energy from the northern hemisphere. This is equivalent to about 100ppm of the total energy received by the northern hemisphere or 1587000 times the energy of the Hiroshima bomb.

This is a just a first estimate ignoring further heat absorbed by the molten water and the increased absorption of solar energy by ocean surface compared to ice and snow.

---

## Short Comment (SC2) · 16 Jan 2019

Based on PIOMAS we estimated the Arctic sea ice loss was 300 km3 per year on average over the past 40 years. By comparison Greenland ice loss was 200 km3 per year on average. Greenland ice volume is two order of magnitude bigger than the Arctic sea ice

---

## Short Comment (SC3) · 20 Jan 2019

As a matter of fact according to GRACE (Gravity Recovery and Climate Experiment) mounted on board a NASA satellite launched in 2002, the Greenland Ice sheet has lost 3600 Gigatonnes of ice between 2002 and 2016. That represents 280 Gigatonnes of ice per year on average (1 gigatonne of ice is sequivalent to 1 km3 of water). It is fascinating to realize that the Arctic Ocean has lost about the same quantity of sea ice than the Greenland ice sheet during the same time period despite the fact the Greenland ice sheet volume is 2 orders of magnitude larger and 3 orders of magnitude thicker than the Arctic sea ice.

---

## Short Comment (SC4) · 22 Jan 2019

Today (Tuesday January 22, 2019) on the first page of the New York Times international edition and under "Signs of future thirst", we can read "the world's roughly 150 000 glaciers, not including the large ice sheets of Greenland and Antarctica, cover about 200 000 square miles of the earth's surface. Over the last four decades they've lost the equivalent of a layer of ice 70 feet thick". Translated in metric units this is equivalent to a loss of about 23m of ice over half a million km2 or 11500 km3 of ice loss. This is precisely the same amount of Arctic sea ice loss based on PIOMAS estimations for the same time period (the past 40 years).

---

## Referee Comment (RC1) · Anonymous Referee #1 · 14 Feb 2019

**Summary**

This paper aims to evaluate Arctic sea-ice volume changes from 1979 until now, using PIOMAS Freezing-Degree Days (FDD) based on ERA Interim surface air temperature reanalysis, and Cryosat-2 sea ice thickness estimates. The authors try trie to investigate causes and consequences of the decline in ice volume over the past decades.

**General comments:**

This paper reads like a blog or review, but it significantly lacks scientific rigor and the methods do not seem sound. There are several inconsistencies and inaccuracies in the paper, listed in the following:

1. L26-27 (and later): „In 2018 the Arctic MYI vanished almost completely for the first time ever over the past 40 years". Surely, the amount of MYI has decreased over the past decades and especially the very old sea ice. But I cannot follow this statement and I think it is wrong. Below are quicklooks of the OSI SAF ice type product from March and December 2018, still showing a significant amount of MYI in the Arctic. The authors refer to the cumulative FDD sea ice thickness-based estimation, which I think is very problematic (see next point).

[Figure]

2. I think FDD are useful for the evaluation of first-year sea ice growth, but in order to estimate pan Arctic sea-ice volume, this approach is too simplistic. The snow cover is not represented realistically, ice dynamics are not taken into account, and as the authors state (L 281-282), the FDD cannot explicitly account for MYI. Of course, the winter ice growth is very much driven by thermodynamics, and therefore, the authors find a rude correlation between the FDD volume and the PIOMAS estimates. However, in order to model pan-Arctic ice volume, this FDD approach is not state of the art.

3. In general the paper lacks crucial information about the applied methods. How is the ice volume calculated? Is it consistent for the three different methods (CryoSat-2, FDD, PIOMAS)? Then there are also a lot of statements without being proofed by either own findings or other studies, e.g. L 70-71, or L 306-312. In the summary (L 465 - 468), the authors state they have "confirmed the prediction of Wang and Overland [2009] for a blue (summer ice-free) Arctic Ocean by 2037 or even earlier (2030-2035)". I cannot find any predictions in the paper.

4. The quality of the figures is very low and sometimes it is impossible to discriminate the different colors and symbols.

Therefore, I suggest to reject the paper.

**Detailed Comments:**

L70-71: Needs a citation. Based on which study?

L 269-271: How do you really compute the volume? The ice extent is a binary value and does not represent the actual ice covered area. Over which area did you calculate the volume? Did you use the same land/ocean masks for CryoSat-2, the FDD, and PIOMAS? All these information are crucial and not given here.

L 273: Why 1% uncertainty for the ice extent? Based on what?

L 281-281: "Consequently FDD cannot explicitly account for MYI": I think that is a major shortcoming here. You shouldn't apply it then for MYI.

L 416-417: I don't see how this is predicted in the paper. And based on the chosen methods, I am not sure if any prediction would be reasonable.

Figures 5, 8 and 10: It is really hard and almost impossible to discriminate all the different colors or symbols.

---

## Author Comment (AC1) · 21 Feb 2019

First of all we would like to respond to the reviewer's first two general comments and a specific comment (comment 4). As we said "FDD cannot explicitly account for MYI". This is due to the fact that we started integration for sea ice growth H=0 at t=0 (line 213). The key word here is "explicitly". But we did show (lines 224 to 227)"a sharp double peak FDD spatial distribution" followed by "a double peak sea ice thickness distribution typical of the Arctic Ocean and related to FYI and MYI"(line 249). That meant the FDD approach is able to implicitly account for MYI, at least partially. This is because surface air temperature (at 2m), the parameter used for calculating FDD, depends on sea ice thickness (i.e. thicker MYI versus thinner FYI spatial distribution) via air-ice-ocean interactions. Initially it was not obvious that FDD would be appearing

spatially like a double peak distribution similar to Arctic FYI and MYI. As a matter of fact the FDD double peak spatial distribution appeared very clearly for all the time series we analyzed extending over the past 40 years except for last year (2018) when only a single peak FDD distribution showed up. This is more likely a tipping point regarding MYI evolution, indicating MYI is vanishing to the point it is getting more and more difficult to distinguish MYI from FYI. The Eumetsat maps representing MYI in March and December 2018 are still questionable. The definition of MYI is quite broad and vague. On many instances MYI is everything else but FYI and it could represent many different things such as second year ice, deformed ice, ridged ice rather than MYI and still be interpreted as old ice. The new fact revealed by the FDD approach for not being able to discriminate MYI from FYI in 2018 for the first time in 40 years is an important result. Of course it does not mean that MYI has completely disappeared (not quite yet). Consequently we are going to reformulate our statement (lines 26-27 and 281-282) in a less controversial way based on the fact that the FDD approach is ringing a bell concerning the MYI vanishing (rather than having disappeared completely). Let us respond quite specifically to all the other important points raised by the reviewer as far as general comment 2 are concerned. According to the reviewer "FDD are useful for the evaluation of FYI growth". We do agree "the snow cover is not represented realistically". The reviewer is right the snow cover is important and we did pay attention to it even if we have not solved the issue thoroughly. We showed how important and influential is a thin (few centimeters) snow layer on top of sea ice (see figure 3 and figure 11).The best fit between FDD and PIOMAS was obtained with a snow layer of 5cm uniformly distributed (figure 11). With better snow layer observations over sea ice we could easily take this snow effect into account in a more realistic way. But we are still lacking a realistic snow cover for the Arctic Ocean unfortunately. "ice dynamics are not taken into account". This is true and there is a justification as demonstrated by Stroeve et al. 2019 (lines 620-621) ice dynamics are contributing much less than thermodynamics for sea ice growth in winter 2016-2017. Based on CICE (the Los Alamos model) results ice dynamics was contributing to +1 to +4 cm for sea ice growth compared to -11 to -13 cm

for thermodynamics according to Stroeve et al. (2019). "of course the winter ice growth is very much driven by thermodynamics". Not only we do agree with the reviewer but the FDD approach is proving this is true. Compared to the 11 to 13 cm thermodynamic ice growth reduction mentioned by Stroeve et al. 2019 based on CICE, the FDD approach gave a very similar results (14 to 15 cm sea ice thickness reduction). This makes us confident about the FDD approach being realistic and not too simplistic. "the authors find a rude correlation between the FDD volume and the PIOMAS estimates". It is true that the FDD approach and PIOMAS sea ice volume estimates were giving very different answers at the beginning of the time series at a time when MYI was abundant. That was 30 to 40 years ago. But that is not the case for more recent years. The similarity between PIOMAS and FDD sea ice volume estimates for the past 10 years is remarkable in particular for matching the large amplitude interannual variability so well. This is not a "rude" correlation. Let us now answer other questions and comments from the reviewer. General comment 3 and specific comments 1 and 2 We will improve the section concerning the way we calculated sea ice volume and we will provide all the necessary "crucial information about the applied methods". We will also provide basic information about the "consistency" for the 3 different methods (Cryosat-2, PIOMAS and FDD) used for estimating sea ice volume without repeating and duplicating what has already been published extensively regarding Cryosat-2 and PIOMAS. Regarding lines 70-71 (this is also specific comment 1) the results are based on PIOMAS and we will reformulate this sentence in a clearer way. Lines 306 -312. We agreed with the anonymous reviewer that lines 306-312 are not necessary (as also mentioned in the text). In order to avoid any confusion and/or misinterpretation we will delete lines 306-312 in a new version of the text. Lines 465-468. The prediction of Wang and Overland (2009) is based on a paper published in GRL entitled "A sea ice free summer Arctic within 30 years". In the abstract of this paper it is indicated "We predict an expected value for a nearly sea ice free Arctic in September by the year 2037". In addition it is also indicated in the same paper for presenting Figure 2 "this provides an expected value for a September nearly sea ice free Arctic in the year 2037". Consequently for

lines 465-468 we suggest to add "nearly" to "sea ice free" and we will delete "or even earlier (2030-2035)". Detailed comment 5 (line 416 – 417). Neither FDD, nor PIOMAS and Cryosat-2 can make real prediction. Consequently we will delete "predict" and will use "envision" instead. General comment 4 and detailed comment 6. We apologize for the bad quality of the figures and we will improve all the figures to make them fully readable. Detailed comment 3 (line 273). We will argue more precisely about the 1% uncertainty for sea ice extent in a new version of the text. In conclusion and responding to the reviewer's comments regarding FDD being "problematic, too simplistic and not state of the art", we believe that simplicity does not necessarily mean simplistic or problematic. The main simplifications we used for the FDD approach concern ocean heat flux and ice dynamics that we considered legitimately less important in winter than conductive heat flux and other influential ice thermodynamic processes such as the snow cover. Simplification is always the case even with the most sophisticated approach. We just tried to demonstrate in this paper that simplifications we used for the FDD approach lead to realistic results. Simplicity might also contribute to the state of the art in science. Many thanks to the anonymous reviewer for spending time and for important and relevant comments concerning our paper "Rapid decline of Arctic sea ice volume" that would help us for improving the quality of the paper. We will certainly be able to improve scientific rigor and the quality of the figures quite significantly.

---

## Author Comment (AC2) · 2 Mar 2019

On Figure 3 of the main text we described different functions relating Freezing Degree Days (FDD) and sea ice thickness (SIT) and in particular a linear relationship and two quadratic relationship with a 5cm snow layer thickness on top of sea ice for one and no snow at all for the other. On figure 4 we described the FDD spatial distribution deduced from ERA Interim 2m altitude air temperature above Arctic sea ice in 2017 and 2018. Based on FDD-SIT relationships we deduced sea ice thickness for each ERA Interim grid cell. Knowing the area of each ERA Interim grid cell, we calculated sea ice volume for each grid cell by multiplying SIT and the area of the grid cell and the total Arctic sea ice volume by adding up each grid cell sea ice volume at any given time. On the new figure we represented the sea ice volume obtained from the

three theoretical FDD-SIT relationships (linear and quadratic with or without a snow layer hs= 5cm or 0) that we compared with PIOMAS sea ice volume estimations for the same period (from September 2017 to May 2018). We noticed that FDD sea ice volume estimations are always less than PIOMAS sea ice volume except for the FDD-Sea Ice thickness (SIT) quadratic relationship without any snow (cyan). This is mainly due to the fact that FDD does not explicitly take MYI into account. At the end of the summer (in September) PIOMAS indicated almost 5000 km3 of sea ice that survived the summer melt and mainly representing the sea ice replenishment for the next year made partially of Multiyear and Second year ice (MYI, SYI). At the start of the winter season FDD sea ice volume estimates are 0 by definition as shown on the new figure. The best fit between PIOMAS (red circles) and FDD based sea ice volume estimations is represented by a FDD-SIT quadratic relationship including a 5cm snow layer on top of sea ice (blue crosses on the new figure). The sea ice volume differences are about 3000 km3 between the two estimates during most of the freezing period (mainly due to MYI and SYI as we already said). The sea ice growth rate is nearly 100 km3 per day for both FDD and PIOMAS as well. This is why we mentioned a high correlation between FDD and PIOMAS sea ice volume estimates in addition to the fact that the interannual variability for sea ice volume based on FDD and PIOMAS was also highly correlated as shown on figure 11 (see the main text) in particular for recent years.

[Figure]

**Fig. 1.** sea ice volume (km3) from September 2017 to May 2018 deduced from FDD and PIOMAS

---

## Referee Comment (RC2) · Anonymous Referee #2 · 20 Mar 2019

Overall: This paper attempts to assesses the causes of the rapid decline of the Arctic sea ice volume. For this task Freezing degree days are used, based on ERA-Interim temperatures, and results for derived ice volume are compared with PIOMAS an Cryosat-2. Unfortunately, I have to recommend that the paper should be rejected. I hesitate to write such a bad review, but unfortunately I feel it is justified for this paper. I started taking detailed notes, but then stopped for the middle part of the paper as it became clear that there is way to much wrong to make this publishable even in a major revision. The paper lacks scientific rigor, both in the analysis technique, which admittedly used the data sets it used for "purely arbitrary" reasons and in it's writing. In terms of the writing, there is a lack of citations where other people's work is described (bordering on plagiarism as many of those results are then listed in the summary under "main

outcomes resulting from this study". A student at a university would be reported for plagiarism for submitting this paper!), hard to follow writing style with unclear structure, and imprecise writing (many in-precise and sometimes hyped terms, such as "dangerously", "seriously predict", "confuse sea ice retrievals"). The methodology is also not at all well described and it absolutely not reproducible. At least half of the "main outcomes resulting from this study" are not supported by the analysis done, and several of them are actually at odds with other research that did actually study it, as well as hyping climate change impacts ("dangerously approaching"), which is not scientific nor justified by the data. The paper in many parts reads like a review, but without proper citations in reporting others people's work. So neither as an original contribution nor as a review do I see merit to publishing this manuscript, especially in a highly regarded journal such as The Cryosphere.

Some specific comments, to illustrate the many things wrong with this paper that led me to conclude that reject is my recommendation:

-Title: Need to take out "consequences", as consequences are not analyzed, just extensively speculated about. But all that needs to go, as none of it is backed up by analysis.

-Line 37/38: This is not correct: "actual Arctic sea ice decline is one of the most representative characteristics of climate change.". Arctic sea ice decline is one of the most obvious effects of climate change. Not the "most representative"

-Line 38/39: "In the past, the main aspect concerned Arctic sea ice extent largely based on space observations. " Incomplete sentence.

-Line 41/42: Missing closing parenthesis "an abrupt decline in the Arctic winter sea ice cover in 2005 and 2006"

-Line 43/45: No citation for the 2007 minimum?

-Line 45/46: I like the IPY as much as you, but this statement needs proof to back it up
or it has to go in a scientific article: "Thanks to the IPY stimulating a major effort from the scientific community, the first decade of the 21st century ended with an unprecedented amount of new results regarding the Arctic sea ice." Are those all attributable to the IPY (Which should be possible to track down based on acknowledgements and IPY publication database). But I don't see why this statement needs to be here in the first place, so I would recommend removing it.

-Line 47/48: Those citations need to go, we have no idea what they are cited for scientifically, apart from being IPY products. No scientific reason for these citations, so please remove them.

-Line 49/50: "Zhao et al. (2018) described a strong decrease of sea ice concentration in the entire central Arctic in 2010." In what season? Annually? Needs more context. Wasn't another minimum.

-Line 50: Please change "The whole time record" to "the absolute record"

-Line 53-55: Unclear sentence, please revise. "This continuous chain of events maintained a strong motivation among scientists for Arctic sea ice research both from a modeling and experimental point of view taking advantage of new technologies for observations and more sophisticated models. "

-Line 56: remove "peculiar and intriguing", it's imprecise, open to interpretation, and not a scientific term

- Line 63: Who did this? "In order to better analyze and understand Arctic sea ice evolution, an important step was accomplished by introducing Arctic sea ice volume."

-Line 78-89: This should be in a methods section. Not the introduction

-Line 91: "In this Introduction": this is not a introduction, it is a methods or results section (not clear). Please revise

-Line 91/92: Needs a reference for PIOMAS right here at first mention. Also needs a

written out name for PIOMAS

-Line 118: These two periods are not "overlapping", so why does it say "overlapping" here?

-Line 140-144: Unclear why this bouncing ball anecdote is included here, it is not referred to again in the following sentence. So should be removed. There are much better papers to reference for internal variability which should be cited instead (Winton 2011, Kay et al 2011, Swart et al 2015 as cited, etc). The bouncing ball is useful to explain internal variability to a lay audience, as done by Ed Hawkins in his blog as quoted, but has no place in a paper in The Cryosphere, read by cryospheric scientists.

-Line 148: Why are the CMIP models able to better reflect the observed variability? Averaging them all mutes the variability and the thickness and area, and hence, volume , in many of them is very wrong (Massonnett et al 2018). Also, better than what? "Averaged projection from 30 CMIP5 models that can better reflect the observed sea ice volume climatology and variability"

-Line 154: Exactly, the CMIP5 models are not very good. Sorry, but none of this makes sense

-Line 158/159: Who introduced the idea of ice volume from FDD? Needs a refence here as well as much, much more information on this method, as it seems very odd. It also doesn't match PIOMAS variability well, as shown by the reply to Review 1 comments. And it is unclear why one would want to use that over PIOMAS in the first place, that case needs to be made first. Using reanalysis temperatures over the Arctic likely includes large errors. What is the point of using this old method? What are the details of the method? Has to be reproductable, so needs a lot more detail.

-Line 166-168: Why focus on that period? Should have a scientific reason as to why.

-Line 169: What was the previous section if not methods?

-Line 175: ERA-Interim needs a citation at first appearance.

-Line 325: not scientific "confuses" sea ice thickness retrievals

-Line 329: We can not publish "purely arbitrary" stuff. The sensitivity to all choices has to be investigated, so that we know if any conclusions are robust.

-Line 332: For simplicity we did not want to open a new section by inter-comparing different data set and co-evaluating various models." Well, for scientific accuracy, and to publish a paper, one has to do at least some assessment of how robust the results are, if the choice of the data sets was "purely arbitrary" rather than following some well thought out plan.

-Line 359: "dangerously" not scientific, please remove

-Line 409: "It seems quite clear" I think there is still significant scientific debate on this point, which needs to be reflected in the paper.

-Line 416/17: "We can seriously predict a quasi disappearance of the Arctic sea ice in summer during the coming decades." How? You didn't do any forward modeling, and didn't even describe extending the trend into the future. So how can you "seriously predict"? And what is "seriously predict" in the first place? "seriously" has no place here, or in any scientific writing.

-Line 433: "blue Arctic" was first used in a scientific paper, as far as I know, in Newton et al (2016). Needs to be cited.

-The fact that the CMIP5 models have a huge spread of ice-free contitions has been dicussed in many, many papers and the IPCC report. You didn't show that, you didn't even look at the CMIP5 models. So you need to cite those studies. Just to name a few: Overland and Wang 2010, Massonett et al 2012, Jahn et al 2016.

-Line 438-441: Several papers have looked at just that, how the sea ice will transition to an ice-free summer Arctic over the 21st century, under different scenarios and with the impact of internal variability. Need to cite those (Sigmond et al. 2018, Jahn 2018, Notz and Stroeve 2018)

-Line 443-457: Not a single reference in the whole paragraph, but it's all on previous work. Needs references.

-Main points of the study

-Line 459: Ice volume was not directly compared with ice extent and thickness, so how did the study show that ice volume "It is more sensitive"? Not supported by the analysis done.

-Line 465-466: No future projections at all, so how was the projection of Wang and Overland 2009 confirmed? It wasn't so this again isn't supported by analysis done.

-Line 466-468: "Considering that today 75% of the Arctic sea ice volume melted during summer, it would not take long to melt away the remaining 25% of Arctic sea ice in summer." Not if it is all linear, but other people's work (Bitz et al 2004 or so) has shown that thin ice actually grows faster, so it's probably not linear. And again, this wasn't shown here in any way, so not supported by analysis.

-Line 469-471: Again, no analysis of CMIP5 models, so this point is also not shown here. It is taken from other's work.

-Line 472-475: Not shown in this paper, and other people's papers actually don't show that, using models and actually looking at an ice-free Arctic. Here what was looked at was the ice volume over the observed record.

-Line 492-293: Exactly, that is one factor that really troubles me about this FDD method, and a sensitivity study is needed before writing a paper on it, to see how it changes the results.

-Line 494-496: Of course the albedo feedback isn't important in the winter, there is no sunlight so albedo doesn't matter. Not really a new insight, even though I don't know who first wrote about that.

-Line 516: "We would like to suggest using more extensively Arctic sea ice volume

deduced from cumulative FDD in particular to evaluate the impact of climate change on Planet Earth in the future. " I can not suggest that, as first you need to show how it varies using different reanalysis products, and assess the effect of snow on sea ice. Also, I don't even think anyone can use it, as the methodology isn't clearly explained, and I have a lot of questions about it.

-Figures: All way to low quality (fuzzy).

- Figure 5: can't tell different years from each other, needs a better color scheme.

---

## Author Comment (AC3) · 26 Mar 2019

Response to Referee 2 Let us first respond to specific comments Title : "Need to take out consequences"? We agree and will delete not only "consequences" but also "causes" in the title. Line 37/38 "Arctic sea ice decline is one of the most obvious effects of climate change" and "not the most representative". We agree. This is what we meant. Line 38/39. "Incomplete sentence". We will delete it or rephrase it. We just meant that space observations were very instrumental for a better understanding of what is going on in the Arctic. Line 41-42 "Missing closing parenthesis". Sorry about that. We will correct it. Line 43-45. "No citation for the 2007 minimum?" Yes we have 4 citations for the 2007 minimum (lines 47 and 48). We will move them from lines 47-48 up to lines 43-45. Line 45-46. The referee suggested to remove the sentence regarding

[Figure]

IPY and the unprecedented amount of new results regarding Arctic sea ice. No problem we will remove this sentence since it was not our intention to make a complete survey of IPY "new results." The referee is right "it has to go in a scientific article" but this was not our main objective. Line 47-48. "Those citations need to go" and "no scientific reason for these citations". This is in contradiction with the question raised line 43-45 by Referee 2 for the 2007 minimum that occurred precisely during the IPY. We could certainly move these citations from line 47-48 up to line 43-45. Line 49-50. Zhao et al (2018) cited in the last reference lines 634-636, mentioned "Record low sea ice concentration in the central Arctic during summer 2010." We could certainly better introduce this citation in the text line 49-50. The word "summer" was missing there. Line 50. Please change "the whole time record" to the "absolute record". OK we will do so. Line 53-55. "unclear sentence". "please revise". We will remove this sentence. It does not bring anything to the text. Line 56. Remove "peculiar and intriguing". OK we will do so Line 63. "Who did this?" In fact we contributed to it but not only us of course and we can provide an extensive list of who contributed in introducing Arctic sea ice volume. Most of these contributors are listed in the references. No problem we will clear up this point. Line 78-89 and line 91. "This is not an introduction". Referee 2 is correct. We will restructure this part of the text separating what is the real introduction from what is something else (such as methodology and results). So we do agree with the referee's comment. Line 91-92. "Needs a reference for PIOMAS right here at first mention, also needs to write out the full name for PIOMAS". PIOMAS was first mentioned in lines 73 -74 together with a reference and the full name. Line 118. These 2 periods are not "overlapping". Referee 2 is right; "overlapping" is not the proper word and we will delete it. Line 140-144. The "bouncing ball anecdote" should be removed. We agree with Referee 2 and we will remove this sentence. "There are much better papers to reference for internal variability" and we do agree. This is why we cited (3 times) Swart et al 2015 since we found it very appropriate and very relevant to our own findings. We will make sure this aspect will be further discussed in the text together with main outcomes in the conclusion. Line 148 "why are the CMIP models

able to better reflect the observed variability" and "better than what?" is not the proper question we addressed. We referred to a citation of Song 2016 "Using the averaged projection of those climate models from the 30 CMIP5 models that can better reflect the observed sea ice volume climatology and variability, it is shown that the September sea ice volume will decrease to 3000 km3 in the early 2060s and then level off under a medium-mitigation scenario". It is not our intention to analyze all the CMIP model results that have been extensively published. Our intention is only to refer to those publications that are relevant for us and our study regarding Arctic sea ice volume estimated by and /or from CMIP models in order to compare it with our own results based on the FDD approach and PIOMAS. As a matter of fact, we found three very relevant papers 1/ "Assessment of sea ice simulations in the CMIP5 models" by Shu et al 2015 in The Cryosphere, 2/ "Change of Arctic sea ice volume and its relationship with sea ice extent in CMIP5 simulations" by Song 2016 in Atmospheric and Oceanic Science Letters and 3/ "Influence of internal variability on Arctic sea-ice trends" by Swart et al 2015 in Nature Climate Change. Line 154 "the CMIP5 models are not very good"and "sorry but none of this makes sense". We are not sure what does that mean? This is Referee's 2 own assessment regarding CMIP5 model results. Of course there are significant differences between CMIP5 estimations and PIOMAS projection. In particular we found a time difference of about 20 years in reaching the 3000km3 sea ice volume minimum described by Song 2016. But we think it is interesting and important to compare PIOMAS results with CMIP5 results as Song started to do it in 2016 (Atmospheric and Oceanic Science Letters Vol 9 NO.1, 22-30). Line 158-159. "Who introduced the idea of ice volume from FDD?" We introduced this idea following also some previous studies such as Maykut (1986). The idea is old (but not "odd") and we applied it to a new data set (ERA Interim). One of our main conclusions is that FDD sea ice based calculations match well PIOMAS sea ice volume in particular when estimating Arctic sea ice growth rate in winter (113 km3 per day deduced from both FDD and PIOMAS as well). Line 166-168. "Why focus on that period?" The period is defined by the ERA Interim reanalysis. From a purely scientific point of view this period

is interesting because it extends over 40 years exhibiting significant changes and also takes advantage of coherent satellite observations starting during the late 70s. Line 169. "What was the previous section if not methods?" We will correct that together with lines 78-89 and line 91 (see previous comments). Line 175. "ERA Interim needs a citation at first appearance." We cited Simmons et al 2004 and Berrisford et al 2011 in line 180. We could move it up to line 175. No problem. Line 325. "confuses" not scientific. OK we will remove it. Line 329. "we cannot publish "purely arbitrary"stuff". We do agree with referee 2 that "purely arbitrary" is not an appropriate word to qualify the choice we made with the FDD approach and a comparison with PIOMAS. This is an option that we can justify. We will rephrase this sentence accordingly. Line 332. "How robust the results are?" As we said our main intention was not (and still is not) to write a paper regarding all the various approaches to estimate Arctic sea ice volume. Our main intention is to compare the FDD approach for estimating sea ice volume with PIOMAS in particular and also with CICE for a special case. We also compared Cryosat-2 in order to have an observational perspective. Note that Cryosat-2 and PIOMAS sea ice volume estimations have been recently published (e.g., Tilling et al 2017). We found interesting to compare both with the FDD approach (figure 12). The robustness of the results is mainly based on 40 years of calculations documenting both the long term trend and the interannual variability. Line 359 "dangerously" not scientific. "Please remove." We agree and we will remove it. Line 409 "It seems quite clear". "There is still significant debate on this point which needs to be reflected in the paper." We do agree with referee 2 comment and will proceed accordingly. Line 416-417. "We can seriously predict". We completely agree with the referee's comment. First of all "seriously" is not an appropriate word in that context and so we will remove it. Second we are not really making "prediction". So this is a double mistake from our side and we thank referee 2 for pointing it out. Line 433 "blue Arctic" was first used in Newton et al 2016. We will remove it. "CMIP5 models have been discussed in many, many papers". Yes this is correct and this is why we selected some of the CMIP5 results related to studies made by Song (2016), Shu et al (2015), Swart et al 2015. See our previous comments

line 148 and line 154. Line 438-441. "several papers have looked at just that" " "need to cite those". Again we did mention Shu et al (2015), Song (2016) and Swart et al (2015) to foresee how the sea ice will transition to an ice free summer Arctic over the 21st century under different scenarios and with the impact of internal variability. As suggested by Referee 2 we will update this aspect by looking at Sigmond et al 2018, Jahn 2018, Notz and Stroeve 2018 that all represent more recent studies (2018) than those we used (2015-2016). We want to make sure our 2015-2016 references are not outdated by most recent ones (2018). Line 443-457. "not a single reference in the whole paragraph". We do agree with the referee. There are obviously missing references in this paragraph. Thank you for bringing our attention to that important point. We will fix it.

Main points of the study Line 459. "Ice volume was not directly compared with ice extent and thickness". What is supported by PIOMAS analysis is a reduction of 75% of sea ice volume at the end of the summer for the most recent period (4000 km3) compared to Arctic sea ice summer during late 70s early 80s (16000 km3). During the same period of time sea ice extent has changed from more than 6 million km2 to about 3 million km2 (that is 50 % reduction for sea ice extent). Implicitly and only based on sea ice volume reduction and sea ice extent reduction, it would imply a sea ice thickness reduction of 50% over the same time period (this is purely mathematical). The word "more sensitive" comes from the fact that we compared a 75% reduction for sea ice volume to a 50% reduction in sea ice extent and in sea ice thickness over the same period of time. We plan to discuss more the sea ice extent and thickness in the revised version of the text in addition to sea ice volume. Line 465-466. "No future projections at all". We will avoid using the word "prediction" since PIOMAS has no real predictive capabilities. But here we are actually making "projections" based on the tendency observed during the past 40 years, including both the long term trend and the interannual variability. Line 466-468. "Bitz et al 2004 showed that thin ice actually grows faster, so it is probably not linear". In fact the notion that thin ice grows faster than thick ice was already well understood during the 8Os (see the geophysics of sea

ice by Untersteiner 1986) and this was largely taken into account again in our study and supported by our analysis. We presented 3 quadratic relationships relating FDD and sea ice growth and one linear relationship to identify the differences and the sensitivity for sea ice volume estimations. It appears that the important effect of the snow layer on sea ice is increasing the importance of the linear terms of the relationships between FDD and sea ice growth as also mentioned by Maykut in 1986. Line 469-471. "Again no analysis of CMIP5 models in this study". It was (and still is) not our intention to analyze CMIP5 models. We took advantage of CMIP5 published results (Song and Shu et al. already mentioned and also Swart et al 2015) in order to compare them with other sea ice volume estimations such as those based on PIOMAS for the past 40 years. The main result is to indicate a lag of about 20 years between the CMIP5 models and PIOMAS in sea ice volume decline. This is not plagiarism. We can again addup the two CMIP5 references (Song and Shu et al) in order to clear up this aspect. Line 472-475. "Not shown in this paper. Here what was looked at was ice volume over the observed period." We did refer to Swart et al 2015 (cited 3 times in our paper) revealing from CMIP5 models a 7 year internal variability that is also appearing from PIOMAS as well during the past 40 years. So we could recall again Swart et al study line 472-475. Line 492-493. The importance of the snow layer for applying the FDD method. We did spend quite a bit of time and text and figures (3 and 11) to explain the importance of the snow layer for applying the FDD method and we demonstrated how sensitive it is. This was also mentioned by referee 1. We do agree with both referees that the snow layer on top of sea ice is a very important aspect. We concluded from our FDD approach that the quadratic relationship linking FDD and sea ice including a 5cm snow layer depth on top of it, was providing the best fit when compared to PIOMAS. Line 494-496. The albedo feedback in winter is negligible. We do agree. Indeed this is not a scoop. Line 516. We can delete this last sentence. Maybe this is premature. We agree with the referee that we need to use different reanalysis product and this is what we just started to do by using ERA 5 instead of Era Interim. We also realized that applying an old method to new data set is triggering lots of questions and comments

and we agree with the fact that time is needed for all of us to get more familiar with this new approach. Sorry again for the bad quality of the figures that we have all reworked and improved. This was also noted by the referee 1. We realized the bad quality of the figures did not help understanding our paper and we do apologize again. General comments "Half of the main outcomes are not supported by the analysis done" according to referee 2. In the main conclusion we identified 12 main outcomes. What are those half unsupported outcomes ? Obviously the two referees have questions about the methodology we used for the FDD approach and this is quite easy for us to rectify. It is so simple and straightforward that we did not think it was necessary to give all the details. But we agreed with both referees to provide all the information about the methodology we used. Regarding referee 2's comment about "the methodology being absolutely not reproducible", we disagree. It is perfectly reproducible and we could easily provide the software we used (additional materials). We also do believe FDD results match PIOMAS variability well. We strongly suspect the bad quality of the figures being instrumental for some misunderstanding expressed by the two referees? We have already improved the quality of the figures. References. It is quite possible that we missed some specific reference at a proper place in the main text and as we already said (lines 443 to 457) we can correct it by inserting more citations. We feel quite uncomfortable with referee 2's comment about "plagiarism". This is a very severe and unjustified accusation. We were able to respond point by point to all the remarks made by Referee 2 as far as those remarks were specific and explicit. We agreed with most of them and we disagreed with few. So there is no need for unjustified and excessive accusation. There is also obviously a question about the language we used (english) since none of us are native english speaking scientists. We apologize for this inconvenience but we ask for the indulgence of the referees and editors since in addition to science, we have to make it in a proper language. Overall, we would like thank referee 2 for his (or her) time and many constructive comments.